# INTERNEURONS ACCELERATE LEARNING DYNAMICS IN RECURRENT NEURAL NETWORKS FOR STATISTICAL ADAPTATION

**David Lipshutz**[1]**, Cengiz Pehlevan**[2]**, Dmitri B. Chklovskii**[1,3]
[1]Center for Computational Neuroscience, Flatiron Institute
[2]John A. Paulson School of Engineering and Applied Sciences, Harvard University
[3]Neuroscience Institute, New York University School of Medicine
`{dlipshutz,dchklovskii}@flatironinstitute.org`
`cpehlevan@seas.harvard.edu`

## ABSTRACT

Early sensory systems in the brain rapidly adapt to fluctuating input statistics, which requires recurrent communication between neurons. Mechanistically, such recurrent communication is often indirect and mediated by local interneurons. In this work, we explore the computational benefits of mediating recurrent communication via interneurons compared with direct recurrent connections. To this end, we consider two mathematically tractable recurrent linear neural networks that statistically whiten their inputs — one with direct recurrent connections and the other with interneurons that mediate recurrent communication. By analyzing the corresponding continuous synaptic dynamics and numerically simulating the networks, we show that the network with interneurons is more robust to initialization than the network with direct recurrent connections in the sense that the convergence time for the synaptic dynamics in the network with interneurons (resp. direct recurrent connections) scales logarithmically (resp. linearly) with the spectrum of their initialization. Our results suggest that interneurons are computationally useful for rapid adaptation to changing input statistics. Interestingly, the network with interneurons is an overparameterized solution of the whitening objective for the network with direct recurrent connections, so our results can be viewed as a recurrent linear neural network analogue of the implicit acceleration phenomenon observed in overparameterized feedforward linear neural networks.

## 1 INTRODUCTION

Efficient coding and redundancy reduction theories of neural coding hypothesize that early sensory systems decorrelate and normalize neural responses to sensory inputs (Barlow, 1961; Laughlin, 1989; Barlow & Földiák, 1989; Simoncelli & Olshausen, 2001; Carandini & Heeger, 2012; Westrick et al., 2016; Chapochnikov et al., 2021), operations closely related to statistical whitening of inputs. Since the input statistics are often in flux due to dynamic environments, this calls for early sensory systems that can rapidly adapt (Wark et al., 2007; Whitmire & Stanley, 2016). Decorrelating neural activities requires recurrent communication between neurons, which is typically indirect and mediated by local interneurons (Christensen et al., 1993; Shepherd et al., 2004). Why do neuronal circuits for statistical adaptation mediate recurrent communication using interneurons, which take up valuable space and metabolic resources, rather than using direct recurrent connections?

A common explanation for why communication between neurons is mediated by interneurons is Dale's principle, which states that each neuron has exclusively inhibitory or excitatory effects on all of its targets (Strata & Harvey, 1999). While Dale's principle provides a physiological constraint that explains why recurrent interactions are mediated by interneurons, we seek a computational principle that can account for using interneurons rather than direct recurrent connections. This perspective is useful for a couple of reasons. First, perhaps Dale's principle is not a hard constraint; see (Saunders et al., 2015; Granger et al., 2020) for results along these lines. In this case, a computational benefit of interneurons would provide a normative explanation for the existence of interneurons to mediate

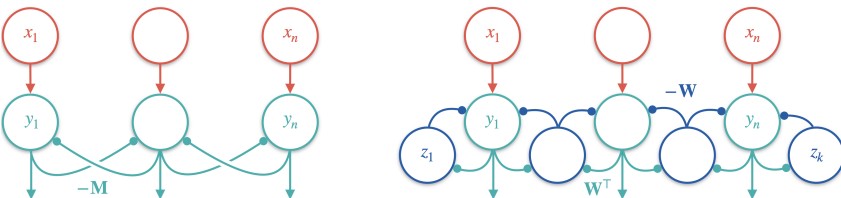

Figure 1: Recurrent neural networks for ZCA whitening with direct recurrent connections (left, Algorithm 1) and with interneurons (right, Algorithm 2).

recurrent communication. Second, decorrelation and whitening are useful in statistical and machine learning methods (Hyvärinen & Oja, 2000; Krizhevsky, 2009), especially recent self-supervised methods (Ermolov et al., 2021; Zbontar et al., 2021; Bardes et al., 2022; Hua et al., 2021), so our analysis is potentially relevant to the design of artificial neural networks, which are not bound by Dale's principle.

In this work, to better understand the computational benefits of interneurons for statistical adaptation, we analyze the learning dynamics of two mathematically tractable recurrent neural networks that statistically whiten their inputs using Hebbian/anti-Hebbian learning rules — one with direct recurrent connections and the other with indirect recurrent interactions mediated by interneurons, Figure 1. We show that the learning dynamics of the network with interneurons are more robust than the learning dynamics of the network with direct recurrent connections. In particular, we prove that the convergence time of the continuum limit of the network with direct lateral connections scales linearly with the spectrum of the initialization, whereas the convergence time of the continuum limit of the network with interneurons scales logarithmically with the spectrum of the initialization. We also numerically test the networks and, consistent with our theoretical results, find that the network with interneurons is more robust to initialization. Our results suggest that interneurons are computationally important for rapid adaptation to fluctuating input statistics.

Our analysis is closely related to analyses of learning dynamics in feedforward linear networks trained using backpropagation (Saxe et al., 2014; Arora et al., 2018; Saxe et al., 2019; Gidel et al., 2019; Tarmoun et al., 2021). The optimization problems for deep linear networks are overparameterizations of linear problems and this overparameterization can accelerate convergence of gradient descent or gradient flow optimization — a phenomenon referred to as *implicit acceleration* (Arora et al., 2018). Our results can be viewed as an analogous phenomenon for gradient flows corresponding to recurrent linear networks trained using Hebbian/anti-Hebbian learning rules. In our setting, the network with interneurons is naturally viewed as an overparameterized solution of the whitening objective for the network with direct recurrent connections. In analogy with the feedforward setting, the interneurons can be viewed as a hidden layer that overparameterizes the optimization problem.

In summary, our main contribution is a theoretical and numerical analysis of the synaptic dynamics of two linear recurrent neural networks for statistical whitening — one with direct lateral connections and one with indirect lateral connections mediated by interneurons. Our analysis shows that the synaptic dynamics converge significantly faster in the network with interneurons than the network with direct lateral connections (logarithmic versus linear convergence times). Our results have potential broader implications: (i) they suggest biological interneurons may facilitate rapid statistical adaptation, see also (Duong et al., 2023); (ii) including interneurons in recurrent neural networks for solving other learning tasks may also accelerate learning; (iii) overparameterized whitening objectives may be useful for developing online self-supervised learning algorithms in machine learning.

## 2 STATISTICAL WHITENING

Let $n \geq 2$ and $\mathbf{x}_1, \ldots, \mathbf{x}_T$ be a sequence of $n$-dimensional centered inputs with positive definite (empirical) covariance matrix $\mathbf{C}_{xx} := \frac{1}{T}\mathbf{X}\mathbf{X}^\top$, where $\mathbf{X} := [\mathbf{x}_1, \ldots, \mathbf{x}_T]$ is the $n \times T$ data matrix of concatenated inputs. The goal of statistical whitening is to linearly transform the inputs so that the $n$-dimensional outputs $\mathbf{y}_1, \ldots, \mathbf{y}_T$ have identity covariance; that is, $\mathbf{C}_{yy} := \frac{1}{T}\mathbf{Y}\mathbf{Y}^\top = \mathbf{I}_n$, where $\mathbf{Y} := [\mathbf{y}_1, \ldots, \mathbf{y}_T]$ is the $n \times T$ data matrix of concatenated outputs.

Statistical whitening is not a unique transformation. For example, if $\mathbf{C}_{yy} = \mathbf{I}_n$, then left multiplication of $\mathbf{Y}$ by any $n \times n$ orthogonal matrix results in another data matrix with identity covariance (Kessy et al., 2018). We focus on Zero-phase Components Analysis (ZCA) whitening (Bell & Sejnowski, 1997), also referred to as Mahalanobis whitening, given by $\mathbf{Y} = \mathbf{C}_{xx}^{-1/2}\mathbf{X}$, which is the whitening transformation that minimizes the mean-squared error between the inputs $\mathbf{X}$ and the outputs $\mathbf{Y}$ (Eldar & Oppenheim, 2003). In other words, it is the unique solution to the objective

$$\underset{\mathbf{Y} \in \mathbb{R}^{n \times T}}{\arg\min} \frac{1}{T} \|\mathbf{Y} - \mathbf{X}\|_{\text{Frob}}^2 \qquad \text{subject to} \qquad \frac{1}{T}\mathbf{Y}\mathbf{Y}^\top = \mathbf{I}_n. \tag{1}$$

The goal of this work is to analyze the learning dynamics of 2 recurrent neural networks that learn to perform ZCA whitening in the online, or streaming, setting using Hebbian/anti-Hebbian learning rules. The derivations of the 2 networks, which we include here for completeness, are closely related to derivations of PCA networks carried out in (Pehlevan & Chklovskii, 2015; Pehlevan et al., 2018). Our main theoretical and numerical results are presented in sections 6 and 7, respectively.

## 3 Objectives for deriving ZCA whitening networks

In this section, we rewrite the ZCA whitening objective 1 to obtain 2 objectives that will be the starting points for deriving our 2 recurrent whitening networks. We first expand the square in equation 1, substitute in with the constraint $\mathbf{Y}\mathbf{Y}^\top = T\mathbf{I}_n$, drop the terms that do not depend on $\mathbf{Y}$ and finally flip the sign to obtain the following objective in terms of the neural activities data matrix $\mathbf{Y}$:

$$\underset{\mathbf{Y} \in \mathbb{R}^{n \times T}}{\max} \frac{2}{T}\text{Tr}(\mathbf{Y}\mathbf{X}^\top) \qquad \text{subject to} \qquad \frac{1}{T}\mathbf{Y}\mathbf{Y}^\top = \mathbf{I}_n.$$

### 3.1 Objective for the network with direct recurrent connections

To derive a network with direct recurrent connections, we introduce the positive definite matrix $\mathbf{M}$ as a Lagrange multiplier to enforce the upper bound $\mathbf{Y}\mathbf{Y}^\top \preceq T\mathbf{I}_n$:

$$\underset{\mathbf{M} \in \mathcal{S}_{++}^n}{\min} \underset{\mathbf{Y} \in \mathbb{R}^{n \times T}}{\max} f(\mathbf{M}, \mathbf{Y}), \tag{2}$$

where $\mathcal{S}_{++}^n$ denotes the set of positive definite $n \times n$ matrices and

$$f(\mathbf{M}, \mathbf{Y}) := \frac{2}{T}\text{Tr}(\mathbf{Y}\mathbf{X}^\top) - \frac{1}{T}\text{Tr}(\mathbf{M}(\mathbf{Y}\mathbf{Y}^\top - T\mathbf{I}_n)).$$

Here, we have interchanged the order of optimization, which is justified because $f(\mathbf{M}, \mathbf{Y})$ is linear in $\mathbf{M}$ and strongly concave in $\mathbf{Y}$, so it satisfies the saddle point property (see Appendix A). The maximization over $\mathbf{Y}$ ensures that the upper bound $\mathbf{Y}\mathbf{Y}^\top \preceq T\mathbf{I}_n$ is in fact saturated, so the whitening constraint holds. The minimax objective equation 2 will be the starting point for our derivation of a statistical whitening network with direct recurrent connections in section 4.

### 3.2 Overparameterized objective for the network with interneurons

To derive a network with $k \geq n$ interneurons, we replace the matrix $\mathbf{M}$ in equation 2 with the overparameterized product $\mathbf{W}\mathbf{W}^\top$, where $\mathbf{W}$ is an $n \times k$ matrix, which yields the minimax objective

$$\underset{\mathbf{W} \in \mathbb{R}^{n \times k}}{\min} \underset{\mathbf{Y} \in \mathbb{R}^{n \times T}}{\max} g(\mathbf{W}, \mathbf{Y}), \tag{3}$$

where

$$g(\mathbf{W}, \mathbf{Y}) := f(\mathbf{W}\mathbf{W}^\top, \mathbf{Y}) = \frac{2}{T}\text{Tr}(\mathbf{Y}\mathbf{X}^\top) - \frac{1}{T}\text{Tr}(\mathbf{W}\mathbf{W}^\top(\mathbf{Y}\mathbf{Y}^\top - T\mathbf{I}_n)).$$

The overparameterized minimax objective equation 3 will be the starting point for our derivation of a statistical whitening network with interneurons in section 5.

## 4 DERIVATION OF A NETWORK WITH DIRECT RECURRENT CONNECTIONS

Starting from the ZCA whitening objective in equation 2, we derive offline and online algorithms and map the online algorithm onto a network with direct recurrent connections. We assume that the neural dynamics operate at faster timescale than the synaptic updates, so we first optimize over the neural activities and then take gradient-descent steps with respect to the synaptic weight matrix.

In the offline setting, at each iteration, we first maximize $f(\mathbf{M}, \mathbf{Y})$ with respect to the neural activities $\mathbf{Y}$ by taking gradient ascent steps until convergence:

$$\mathbf{Y} \leftarrow \mathbf{Y} + \gamma(\mathbf{X} - \mathbf{M}\mathbf{Y}) \qquad \Rightarrow \qquad \mathbf{Y} = \mathbf{M}^{-1}\mathbf{X},$$

where $\gamma > 0$ is a small constant. After the neural activities converge, we minimize $f(\mathbf{M}, \mathbf{Y})$ with respect to $\mathbf{M}$ by taking a gradient descent step with respect to $\mathbf{M}$:

$$\mathbf{M} \leftarrow \mathbf{M} + \eta \left( \frac{1}{T} \mathbf{Y}\mathbf{Y}^\top - \mathbf{I}_n \right). \tag{4}$$

In the online setting, at each time step $t$, we only have access to the input $\mathbf{x}_t$ rather than the whole dataset $\mathbf{X}$. In this case, we first optimize $f(\mathbf{M}, \mathbf{Y})$ with respect to the neural activities vector $\mathbf{y}_t$ by taking gradient steps until convergence:

$$\mathbf{y}_t \leftarrow \mathbf{y}_t + \gamma(\mathbf{x}_t - \mathbf{M}\mathbf{y}_t) \qquad \Rightarrow \qquad \mathbf{y}_t = \mathbf{M}^{-1}\mathbf{x}_t.$$

After the neural activities converge, the synaptic weight matrix $\mathbf{M}$ is updated by taking a stochastic gradient descent step:

$$\mathbf{M} \leftarrow \mathbf{M} + \eta(\mathbf{y}_t\mathbf{y}_t^\top - \mathbf{I}_n).$$

This results in Algorithm 1.

---

**Algorithm 1:** A whitening network with direct recurrent connections

> **input** centered inputs $\{\mathbf{x}_t\}$; parameters $\gamma, \eta$
> **initialize** positive definite $n \times n$ matrix $\mathbf{M}$
> **for** $t = 1, \ldots, T$ **do**
>     $\mathbf{y}_t \leftarrow \mathbf{0}$
>     **repeat**
>         $\mathbf{y}_t \leftarrow \mathbf{y}_t + \gamma(\mathbf{x}_t - \mathbf{M}\mathbf{y}_t)$
>     **until** convergence
>     $\mathbf{M} \leftarrow \mathbf{M} + \eta(\mathbf{y}_t\mathbf{y}_t^\top - \mathbf{I}_n)$
> **end for**

---

Algorithm 1 can be implemented in a network with $n$ principal neurons with direct recurrent connections $-\mathbf{M}$, Figure 1 (left). At each time step $t$, the external input to the principal neurons is $\mathbf{x}_t$, the output of the principal neurons is $\mathbf{y}_t$ and the recurrent input to the principal neurons is $-\mathbf{M}\mathbf{y}_t$. Therefore, the total input to the principal neurons is encoded in the $n$-dimensional vector $\mathbf{x}_t - \mathbf{M}\mathbf{y}_t$. The neural outputs are updated according to the neural dynamics in Algorithm 1 until they converge at $\mathbf{y}_t = \mathbf{M}^{-1}\mathbf{x}_t$. After the neural activities converge, the synaptic weight matrix $\mathbf{M}$ is updated according to Algorithm 1.

## 5 DERIVATION OF A NETWORK WITH INTERNEURONS

Starting from the overparameterized ZCA whitening objective in equation 3, we derive offline and online algorithms and we map the online algorithm onto a network with $k \geq n$ interneurons. As in the last section, we assume that the neural dynamics operate at a faster timescale than the synaptic updates.

In the offline setting, at each iteration, we first maximize $g(\mathbf{W}, \mathbf{Y})$ with respect to the neural activities $\mathbf{Y}$ by taking gradient steps until convergence:

$$\mathbf{Y} \leftarrow \mathbf{Y} + \gamma(\mathbf{X} - \mathbf{W}\mathbf{W}^\top\mathbf{Y}) \qquad \Rightarrow \qquad \mathbf{Y} = (\mathbf{W}\mathbf{W}^\top)^{-1}\mathbf{X}.$$

After the neural activities converge, we minimize $g(\mathbf{W}, \mathbf{Y})$ with respect to $\mathbf{W}$ by taking a gradient descent step with respect to $\mathbf{W}$:

$$\mathbf{W} \leftarrow \mathbf{W} + \eta \left( \frac{1}{T} \mathbf{Y}\mathbf{Y}^\top \mathbf{W} - \mathbf{W} \right). \tag{5}$$

In the online setting, at each time step $t$, we first optimize $g(\mathbf{W}, \mathbf{Y})$ with respect to the neural activities vectors $\mathbf{y}_t$ by running the gradient ascent-descent neural dynamics:

$$\mathbf{y}_t \leftarrow \mathbf{y}_t + \gamma(\mathbf{x}_t - \mathbf{W}\mathbf{W}^\top \mathbf{y}_t) \qquad \Rightarrow \qquad \mathbf{y}_t = (\mathbf{W}\mathbf{W}^\top)^{-1}\mathbf{x}_t.$$

After convergence of the neural activities, the matrix $\mathbf{W}$ is updated by taking a stochastic gradient descent step:

$$\mathbf{W} \leftarrow \mathbf{W} + \eta(\mathbf{y}_t\mathbf{y}_t^\top \mathbf{W} - \mathbf{W}).$$

To implement the online algorithm in a recurrent neural network with interneurons, we let $\mathbf{z}_t := \mathbf{W}^\top \mathbf{y}_t$ denote the $k$-dimensional vector of interneuron activities at time $t$. After substituting into the neural dynamics and the synaptic update rules, we obtain Algorithm 2.

---

**Algorithm 2:** A whitening network with interneurons

> **input** centered inputs $\{\mathbf{x}_t\}$; $k \geq n$; parameters $\gamma, \eta$
> **initialize** full rank $n \times k$ matrix $\mathbf{W}$
> **for** $t = 1, \ldots, T$ **do**
>     $\mathbf{y}_t \leftarrow \mathbf{0}$
>     **repeat**
>         $\mathbf{z}_t \leftarrow \mathbf{W}^\top \mathbf{y}_t$
>         $\mathbf{y}_t \leftarrow \mathbf{y}_t + \gamma(\mathbf{x}_t - \mathbf{W}\mathbf{z}_t)$
>     **until** convergence
>     $\mathbf{W} \leftarrow \mathbf{W} + \eta(\mathbf{y}_t\mathbf{z}_t^\top - \mathbf{W})$
> **end for**

---

Algorithm 2 can be implemented in a network with $n$ principal neurons and $k$ interneurons, Figure 1 (right). The principal neurons (resp. interneurons) are connected to the interneurons (resp. principal neurons) via the synaptic weight matrix $\mathbf{W}^\top$ (resp. $-\mathbf{W}$). At each time step $t$, the external input to the principal neurons is $\mathbf{x}_t$, the activity of the principal neurons is $\mathbf{y}_t$, the activity of the interneurons is $\mathbf{z}_t = \mathbf{W}^\top \mathbf{y}_t$ and the recurrent input to the principal neurons is $-\mathbf{W}\mathbf{z}_t$. The neural activities are updated according to the neural dynamics in Algorithm 2 until they converge to $\mathbf{y}_t = (\mathbf{W}\mathbf{W}^\top)^{-1}\mathbf{x}_t$ and $\mathbf{z}_t = \mathbf{W}^\top \mathbf{y}_t$. After the neural activities converge, the synaptic weight matrix $\mathbf{W}$ is updated according to Algorithm 2.

Here, the principal neuron-to-interneuron weight matrix $\mathbf{W}^\top$ is the negative transpose of the interneuron-to-principal neuron weight matrix $-\mathbf{W}$, Figure 1 (right). In general, enforcing such symmetry is not biologically plausible and commonly referred to as the weight transport problem. In addition, we do not sign-constrain the weights, so the network can violate Dale's principle. In Appendix B, we modify the algorithm to be more biologically realistic and we map the modified algorithm onto the vertebrate olfactory bulb and show that the algorithm is consistent with several experimental observations.

## 6    ANALYSES OF CONTINUOUS SYNAPTIC DYNAMICS

We now present our main theoretical results on the convergence of the corresponding continuous synaptic dynamics for $\mathbf{M}$ and $\mathbf{W}$. We first show that the synaptic updates are naturally viewed as (stochastic) gradient descent algorithms. We then analyze the corresponding continuous gradient flows. Detailed proofs of our results are provided in Appendix C

### 6.1    GRADIENT DESCENT ALGORITHMS

We first show that the offline and online synaptic dynamics are naturally viewed as gradient descent and stochastic gradient descent algorithms for minimizing whitening objectives. Let $U(\mathbf{M})$ be the

convex function defined by

$$U(\mathbf{M}) := \max_{\mathbf{Y} \in \mathbb{R}^{n \times T}} f(\mathbf{M}, \mathbf{Y}) = \mathrm{Tr}\left(\mathbf{M}^{-1}\mathbf{C}_{xx} - \mathbf{M}\right).$$

Substituting the optimal neural activities $\mathbf{Y} = \mathbf{M}^{-1}\mathbf{X}$ into the offline update in equation 4, we see that the offline algorithm is a gradient descent algorithm for minimizing $U(\mathbf{M})$:

$$\mathbf{M} \leftarrow \mathbf{M} + \eta\left(\mathbf{M}^{-1}\mathbf{C}_{xx}\mathbf{M}^{-1} - \mathbf{I}_n\right) = \mathbf{M} - \eta\nabla U(\mathbf{M}).$$

Similarly, Algorithm 1 is a stochastic gradient descent algorithm for minimizing $U(\mathbf{M})$.

Next, let $V(\mathbf{W})$ be the nonconvex function defined by

$$V(\mathbf{W}) := \max_{\mathbf{Y} \in \mathbb{R}^{n \times T}} g(\mathbf{W}, \mathbf{Y}) = \mathrm{Tr}\left((\mathbf{W}\mathbf{W}^\top)^{-1}\mathbf{C}_{xx} - \mathbf{W}\mathbf{W}^\top\right).$$

Again, substituting the optimal neural activities $\mathbf{Y} = (\mathbf{W}\mathbf{W}^\top)^{-1}\mathbf{X}$ into the offline update in equation 5, we see the offline algorithm is a gradient descent algorithm for minimizing $V(\mathbf{W})$:

$$\mathbf{W} \leftarrow \mathbf{W} + \eta\left((\mathbf{W}\mathbf{W}^\top)^{-1}\mathbf{C}_{xx}(\mathbf{W}\mathbf{W}^\top)^{-1}\mathbf{W} - \mathbf{W}\right) = \mathbf{W} - \frac{\eta}{2}\nabla V(\mathbf{W}).$$

Similarly, Algorithm 2 is a stochastic gradient descent algorithm for minimizing $V(\mathbf{W})$.

A common approach for studying stochastic gradient descent algorithms is to analyze the corresponding continuous *gradient flows* (Saxe et al., 2014; 2019; Tarmoun et al., 2021), which are more mathematically tractable and are useful approximations of the average behavior of the stochastic gradient descent dynamics when the step size is small. In the remainder of this section, we analyze and compare the continuous gradient flows associated with Algorithms 1 and 2.

To further facilitate the analysis, we consider so-called 'spectral initializations' that commute with $\mathbf{C}_{xx}$ (Saxe et al., 2014; 2019; Gidel et al., 2019; Tarmoun et al., 2021). Specifically, we say that $\mathbf{A}_0 \in \mathcal{S}_{++}^n$ is a *spectral initialization* if $\mathbf{A}_0 = \mathbf{U}_x\mathrm{diag}(\sigma_1, \ldots, \sigma_n)\mathbf{U}_x^\top$, where $\mathbf{U}_x$ is the $n \times n$ orthogonal matrix of eigenvectors of $\mathbf{C}_{xx}$ and $\sigma_1, \ldots, \sigma_n > 0$. To characterize the convergence rates of the gradient flows, we define the Lyapunov function

$$\ell(\mathbf{A}) := \|\mathbf{C}_{xx} - \mathbf{A}^2\|_{\mathrm{Frob}}, \qquad \mathbf{A} \in \mathcal{S}_{++}^n.$$

## 6.2 GRADIENT FLOW ANALYSIS OF ALGORITHM 1

The gradient flow of $U(\mathbf{M})$ is given by

$$\frac{d\mathbf{M}(t)}{dt} = -\nabla U(\mathbf{M}(t)) = \mathbf{M}(t)^{-1}\mathbf{C}_{xx}\mathbf{M}(t)^{-1} - \mathbf{I}_n. \tag{6}$$

To analyze solutions of $\mathbf{M}(t)$, we focus on spectral intializations.

**Lemma 1.** *Suppose $\mathbf{M}_0$ is a spectral initialization. Then the solution $\mathbf{M}(t)$ of the ODE 6 is of the form $\mathbf{M}(t) = \mathbf{U}_x diag(\sigma_1(t), \ldots, \sigma_n(t))\mathbf{U}_x^\top$ where $\sigma_1(t), \ldots, \sigma_n(t)$, are the solutions of the ODE*

$$\frac{d\sigma_i(t)}{dt} = \frac{\lambda_i^2}{\sigma_i(t)^2} - 1, \qquad i = 1, \ldots, n. \tag{7}$$

*Consequently,*

$$\frac{d}{dt}(\sigma_i(t)^2 - \lambda_i^2)^2 = -\frac{4}{\sigma_i(t)}(\sigma_i(t)^2 - \lambda_i^2)^2, \qquad i = 1, \ldots, n. \tag{8}$$

From Lemma 1, we see that for a spectral initialization with $\sigma_i(0) \le \lambda_i$, the dynamics of $\sigma_i(t)$ satisfy

$$\frac{d}{dt}(\sigma_i(t)^2 - \lambda_i^2)^2 \le -\frac{4}{\lambda_i}(\sigma_i(t)^2 - \lambda_i^2)^2.$$

It follows that $\sigma_i(t)^2$ converges to $\lambda_i^2$ exponentially with convergence rate greater than $2/\sqrt{\lambda_i}$. On the other hand, suppose $\sigma_i(0) \gg \lambda_i$. From equation 7, we see that while $\sigma_i(t) \gg \lambda_i$, $\sigma_i(t)$ decays at approximately unit rate; that is,

$$\frac{d\sigma_i(t)}{dt} \approx -1. \tag{9}$$

Therefore, the time for $\sigma_i(t)$ to converge to $\lambda_i$ grows linearly with $\sigma_i(0)$. We make these statements precise in the following proposition.

**Proposition 1.** *Suppose $\mathbf{M}_0$ is a spectral initialization and let $\mathbf{M}(t)$ denote the solution of the ODE 6 starting from $\mathbf{M}_0$. If $\sigma_i \leq \lambda_i$ for all $i = 1, \ldots, n$, then for $\epsilon < \ell(\mathbf{M}_0)$,*

$$\min\{t \geq 0 : \ell(\mathbf{M}(t)) \leq \epsilon\} \leq \frac{\sqrt{\lambda_{max}}}{2} \log\left(\ell(\mathbf{M}_0)\epsilon^{-1}\right), \tag{10}$$

*where $\lambda_{max} := \max_i \lambda_i$. On the other hand, if $\sigma_i > \lambda_i$ for some $i = 1, \ldots, n$, then for $\epsilon < \sigma_i^2 - \lambda_i^2$,*

$$\min\{t \geq 0 : \ell(\mathbf{M}(t)) \leq \epsilon\} \geq \sigma_i - \sqrt{\lambda_i^2 + \epsilon}. \tag{11}$$

### 6.3 Gradient flow analysis of Algorithm 2

The gradient flow of the overparameterized cost $V(\mathbf{W})$ is given by

$$\frac{d\mathbf{W}(t)}{dt} = -\frac{1}{2}\nabla V(\mathbf{W}) = \left(\mathbf{W}(t)\mathbf{W}(t)^\top\right)^{-1} \mathbf{C}_{xx} \left(\mathbf{W}(t)\mathbf{W}(t)^\top\right)^{-1} \mathbf{W}(t) - \mathbf{W}(t). \tag{12}$$

Next, we show that $\ell(\mathbf{W}(t)\mathbf{W}(t)^\top)$ converges to zero exponentially for any initialization $\mathbf{W}_0$.

**Proposition 2.** *Let $\mathbf{W}(t)$ denote the solution of the ODE 12 starting from any $\mathbf{W}_0 \in \mathbb{R}^{n \times k}$. Let $\epsilon < \ell(\mathbf{W}_0\mathbf{W}_0^\top)$. For spectral initializations $\mathbf{W}_0\mathbf{W}_0^\top$,*

$$\min\{t \geq 0 : \ell(\mathbf{W}(t)\mathbf{W}(t)^\top) \leq \epsilon\} = \frac{1}{4} \log\left(\ell(\mathbf{W}_0\mathbf{W}_0^\top)\epsilon^{-1}\right). \tag{13}$$

*For general initializations $\mathbf{W}_0\mathbf{W}_0^\top$,*

$$\min\{t \geq 0 : \ell(\mathbf{W}(t)\mathbf{W}(t)^\top) \leq \epsilon\} \leq \frac{1}{2} \log\left(\ell(\mathbf{W}_0\mathbf{W}_0^\top)\epsilon^{-1}\right). \tag{14}$$

## 7 Numerical experiments

In this section, we numerically test the offline and online algorithms on synthetic datasets. Let

$$\text{Whitening error}(t) = \|\mathbf{A}_t^{-1}\mathbf{C}_{xx}\mathbf{A}_t^{-1} - \mathbf{I}_n\|_{\text{Frob}}, \qquad \mathbf{A}_t \in \{\mathbf{M}_t, \mathbf{W}_t\mathbf{W}_t^\top\},$$

where $\mathbf{A}_t$ is the value of matrix after the $t^{\text{th}}$ iterate. To quantify the convergence time, define

$$\text{Convergence time} := \min\{t \geq 1 : \text{Whitening error}(t) < 0.1\}.$$

In plots with multiple runs, lines and shaded regions respectively denote the means and 95% confidence intervals over 10 runs.

### 7.1 Offline algorithms

Let $n = 5$, $k = 10$ and $T = 10^5$. We generate a data matrix $\mathbf{X}$ with i.i.d. entries $x_{t,i}$ chosen uniformly from the interval $(0, 12)$. The eigenvalues of $\mathbf{C}_{xx}$ are $\{24.01, 16.42, 10.45, 6.59, 3.28\}$. We initialize $\mathbf{W}_0 = \mathbf{Q}\sqrt{\alpha}\mathbf{\Sigma}\mathbf{P}^\top$, where $\alpha > 0$, $\mathbf{\Sigma} = \text{diag}(5, 4, 3, 2, 1)$, $\mathbf{Q}$ is an $n \times n$ orthogonal matrix and $\mathbf{P}$ is a random $k \times n$ matrix with orthonormal column vectors. We set $\mathbf{M}_0 = \mathbf{W}_0\mathbf{W}_0^\top = \alpha\mathbf{Q}\mathbf{\Sigma}^2\mathbf{Q}^\top$. We consider spectral initializations (i.e., $\mathbf{Q} = \mathbf{U}_x$) and nonspectral initializations (i.e., $\mathbf{Q}$ is a random orthogonal matrix). We use step size $\eta = 10^{-3}$.

The results of running the offline algorithms for $\alpha = 0.1, 1, 10$ are shown in Figure 2. Consistent with Propositions 1 and 2, the network with direct recurrent connections convergences slowly for 'large' $\alpha$, whereas the network with interneurons converges exponentially for all $\alpha$. Further, as predicted by equation 9, when the eigenvalues of $\mathbf{M}$ are 'large', i.e., $\sigma_i \gg \lambda_i$, they decay linearly.

In Figure 3, we plot the convergence times of the offline algorithms with spectral and nonspectral initializations, for $\alpha = 1, 2, \ldots, 20$. Consistent with our analysis of the gradient flows, the convergence times for network with direct lateral connections (resp. interneurons) grows linearly (resp. logarithmically) with $\alpha$.

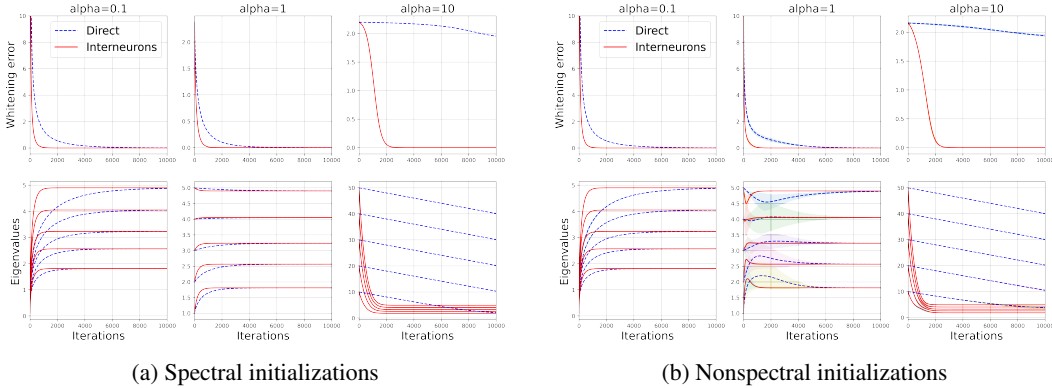

(a) Spectral initializations            (b) Nonspectral initializations

Figure 2: Comparison of whitening error and eigenvalue evolution for the offline algorithms with (a) spectral initializations and (b) nonspectral initializaitons.

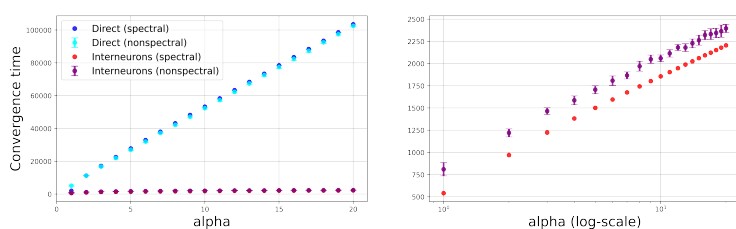

Figure 3: Comparison of convergence times for the offline algorithms with both spectral and non-spectral initializations, as functions of $\alpha = 1, 2, \ldots, 20$.

## 7.2 ONLINE ALGORITHMS

We test our online algorithms on a synthetic dataset with samples from 2 distributions. Let $n = 2$, $k = 4$ and $T = 10^5$. Fix covariance matrices $\mathbf{C}_A = \mathbf{U}_A \mathrm{diag}(4, 25) \mathbf{U}_A^\top$ and $\mathbf{C}_B = \mathbf{U}_B \mathrm{diag}(9, 16) \mathbf{U}_B^\top$, where $\mathbf{U}_A$, $\mathbf{U}_B$ are random $2 \times 2$ rotation matrices. We generate a $2 \times 4T$ dataset $\mathbf{X} = [\mathbf{x}_1, \ldots, \mathbf{x}_{4T}]$ with independent samples $\mathbf{x}_1, \ldots, \mathbf{x}_T, \mathbf{x}_{2T+1}, \ldots, \mathbf{x}_{3T} \sim \mathcal{N}(\mathbf{0}, \mathbf{C}_A)$ and $\mathbf{x}_{T+1}, \ldots, \mathbf{x}_{2T}, \mathbf{x}_{3T+1}, \ldots, \mathbf{x}_{4T} \sim \mathcal{N}(\mathbf{0}, \mathbf{C}_B)$. We evaluated our online algorithms with step size $\eta = 10^{-4}$ and initializations $\mathbf{M}_0 = \mathbf{W}_0 \mathbf{W}_0^\top$, where $\mathbf{W}_0 = \mathbf{Q}\mathrm{diag}(\sigma_1, \sigma_2)\mathbf{P}^\top$, $\mathbf{Q}$ is a random $2 \times 2$ rotation matrix, $\mathbf{P}$ is a random $4 \times 2$ matrix with orthonormal column vectors and $\sigma_1, \sigma_2$ are independent random variables chosen uniformly from the interval $(1, 1.5)$. The results are shown in Figure 4. Consistent with our theoretical analyses, we see that the network with interneurons adapts to changing distributions faster than the network with direct recurrent connections.

## 7.3 APPLICATION TO PRINCIPAL SUBSPACE LEARNING

A challenge in unsupervised and self-supervised learning is preventing *collapse* (i.e., degenerate solutions). A recent approach in self-supervised learning is to decorrelate or whiten the feature

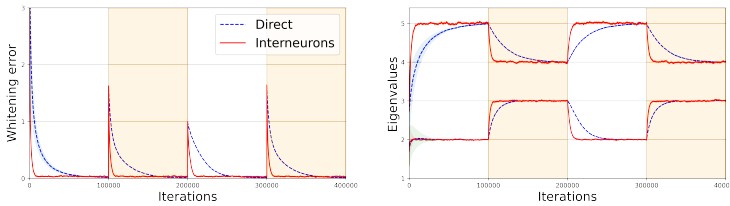

Figure 4: Comparison of whitening error and eigenvalue evolution for the online algorithms on a dataset with switching distributions (white vs. light orange backgrounds).

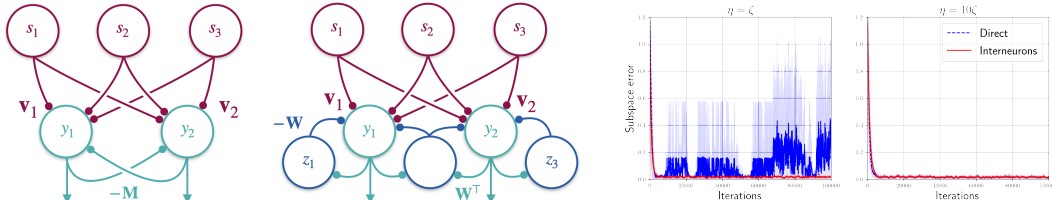

Figure 5: Left: networks for principal subspace projection with output whitening using direct lateral connections or interneurons. Right: comparison of subspace error for the two networks with relative learning rates $\eta = \zeta$ or $\eta = 10\zeta$.

representation (Ermolov et al., 2021; Zbontar et al., 2021; Bardes et al., 2022; Hua et al., 2021). Here, using Oja's online principal component algorithm (Oja, 1982) as a tractable example, we demonstrate that the *speed* of whitening can affect the *accuracy* of the learned representation.

Consider a neuron whose input and output at time $t$ are respectively $\mathbf{s}_t \in \mathbb{R}^d$ and $y_t := \mathbf{v}^\top \mathbf{s}_t$, where $\mathbf{v} \in \mathbb{R}^d$ represents the synaptic weights connecting the inputs to the neuron. Oja's algorithm learns the top principal component of the inputs by updating the vector $\mathbf{v}$ as follows:

$$\mathbf{v} \leftarrow \mathbf{v} + \zeta(y_t \mathbf{s}_t - y_t^2 \mathbf{v}), \tag{15}$$

where $\zeta > 0$ is the step size.

Next, consider a population of $2 \le n \le d$ neurons with outputs $\mathbf{y}_t \in \mathbb{R}^n$ and feedforward synaptic weight vectors $\mathbf{v}_1, \dots, \mathbf{v}_n \in \mathbb{R}^d$ connecting the inputs $\mathbf{s}_t$ to the $n$ neurons. The goal is to project the inputs onto their $n$-dimensional principal subspace. Running $n$ instances of Oja's algorithm in parallel without lateral connections results in *collapse* — each synaptic weight vector $\mathbf{v}_i$ converges to the top principal component. One way to avoid collapse is to whiten the output $\mathbf{y}_t$ using recurrent connections, Figure 5 (left). Here we show that when the subspace projection and output whitening are learned simultaneously, it is critical that the whitening transformation is learned sufficiently fast.

We set $d = 3$, $n = 2$ and generate i.i.d. inputs $\mathbf{s}_t \sim \mathcal{N}(\mathbf{0}, \mathrm{diag}(5, 2, 1))$. We initialize two random vectors $\mathbf{v}_1, \mathbf{v}_2 \in \mathbb{R}^3$ with independent $\mathcal{N}(0, 1)$ entries. At each time step $t$, we use the projection $\mathbf{x}_t := (\mathbf{v}_1^\top \mathbf{s}_t, \mathbf{v}_2^\top \mathbf{s}_t)$ as the input to either Algorithm 1 or 2 and we let $\mathbf{y}_t$ be the output; that is, $\mathbf{y}_t = \mathbf{M}^{-1}\mathbf{x}_t$ or $\mathbf{y}_t = (\mathbf{W}\mathbf{W}^\top)^{-1}\mathbf{x}_t$. For $i = 1, 2$, we update $\mathbf{v}_i$ according to equation 15 with $\zeta = 10^{-3}$ and with $\mathbf{v}_i$ (resp. $y_{t,i}$) in place of $\mathbf{v}$ (resp. $y_t$). We update the recurrent weights $\mathbf{M}$ or $\mathbf{W}$ according to Algorithm 1 or 2. To measure the performance, we define

$$\text{Subspace error} := \|\mathbf{V}(\mathbf{V}^\top \mathbf{V})^{-1}\mathbf{V}^\top - \mathrm{diag}(1, 1, 0)\|_{\mathrm{Frob}}^2, \qquad \mathbf{V} := [\mathbf{v}_1, \mathbf{v}_2] \in \mathbb{R}^{3 \times 2},$$

which is equal to zero when $\mathbf{v}_1, \mathbf{v}_2$ span the 2-dimensional principal subspace of the inputs. We plot the subspace error for Algorithms 1 and 2 using learning rates $\eta = \zeta$ and $\eta = 10\zeta$, Figure 5 (right). The results suggest that in order to learn the correct subspace, the whitening transformation must be learned sufficiently fast relative to the feedforward vectors $\mathbf{v}_i$; for additional results along these lines, see (Pehlevan et al., 2018, section 6). Therefore, since interneurons accelerate learning of the whitening transform, they are useful for accurate or optimal representation learning.

## 8 DISCUSSION

We analyzed the gradient flow dynamics of 2 recurrent neural networks for ZCA whitening — one with direct recurrent connections and one with interneurons. For spectral initializations we can analytically estimate the convergence time for both gradient flows. For nonspectral initializations, we show numerically that the convergence times are close to the spectral initializations with the same initial spectrum. Our results show that the recurrent neural network with interneurons is more robust to initialization.

An interesting question is whether including interneurons in other classes of recurrent neural networks also accelerates the learning dynamics of those networks. We hypothesize that interneurons accelerate learning dynamics when the objective for the network with interneurons can be viewed as an overparameterization of the objective for the recurrent neural network with direct connections.

ACKNOWLEDGMENTS

We thank Yanis Bahroun, Nikolai Chapochnikov, Lyndon Duong, Johannes Friedrich, Siavash Golkar, Jason Moore and Tiberiu Teşileanu for helpful feedback on an earlier draft of this work. C. Pehlevan acknowledges support from the Intel Corporation.

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

## A  SADDLE POINT PROPERTY

Here we recall the following minimax property for a function that satisfies the saddle point property (Boyd & Vandenberghe, 2004, section 5.4).

**Theorem 1.** *Let $V \subseteq \mathbb{R}^n$, $W \subseteq \mathbb{R}^m$ and $f : V \times W \to \mathbb{R}$. Suppose $f$ satisfies the saddle point property; that is, there exists $(\mathbf{a}^*, \mathbf{b}^*) \in V \times W$ such that*

$$f(\mathbf{a}^*, \mathbf{b}) \leq f(\mathbf{a}^*, \mathbf{b}^*) \leq f(\mathbf{a}, \mathbf{b}^*), \qquad \text{for all } (\mathbf{a}, \mathbf{b}) \in V \times W.$$

*Then*

$$\min_{\mathbf{a} \in V} \max_{\mathbf{b} \in W} f(\mathbf{a}, \mathbf{b}) = \max_{\mathbf{b} \in W} \min_{\mathbf{a} \in V} f(\mathbf{a}, \mathbf{b}) = f(\mathbf{a}^*, \mathbf{b}^*).$$

## B  RELATION TO NEURONAL CIRCUITS

In this section, we modify Algorithm 2 to satisfy additional biological constraints and we map the modified algorithm onto the vertebrate olfactory bulb.

### B.1  DECOUPLING THE INTERNEURON SYNAPSES

The neural circuit implementation of Algorithm 2 requires that the principal neuron-to-interneuron synaptic weight matrix $\mathbf{W}^\top$ is the negative transpose of the interneuron-to-principal neuron synaptic weight matrix $-\mathbf{W}$. In general, enforcing this symmetry is not biologically plausible, and is commonly referred to as the weight transport problem. Here, following (Golkar et al., 2020, appendix D), we decouple the synapses and show that the (asymptotic) symmetry of the synaptic weight matrices follows from the symmetry of the local Hebbian/anti-Hebbian updates.

We begin by replacing $\mathbf{W}$ and $\mathbf{W}^\top$ in Algorithm 2 with $\mathbf{W}_{zy}$ and $\mathbf{W}_{yz}$, respectively. Then the neural activities are given by

$$\mathbf{y}_t \leftarrow \mathbf{y}_t + \gamma(\mathbf{x}_t - \mathbf{W}_{zy}\mathbf{z}_t), \qquad\qquad \mathbf{z}_t \leftarrow \mathbf{z}_t + \gamma(\mathbf{W}_{yz}\mathbf{y}_t - \mathbf{z}_t), \qquad (16)$$

which converge to $(\mathbf{W}_{zy}\mathbf{W}_{yz})^{-1}\mathbf{x}_t$ and $\mathbf{z}_t = \mathbf{W}_{yz}\mathbf{y}_t$. After the neural activities converge, the synaptic weights are updated according to the update rules

$$\mathbf{W}_{zy} \leftarrow \mathbf{W}_{zy} + \eta(\mathbf{y}_t\mathbf{z}_t^\top - \mathbf{W}_{zy})$$
$$\mathbf{W}_{yz} \leftarrow \mathbf{W}_{yz} + \eta(\mathbf{z}_t\mathbf{y}_t^\top - \mathbf{W}_{yz}).$$

Let $\mathbf{W}_{zy,t}$ and $\mathbf{W}_{yz,t}$ denote the values of the synaptic weight matrices after $t$ updates. By iterating the above updates, we see that the difference between the weight matrices after $t$ updates is given by

$$\mathbf{W}_{yz,t} - \mathbf{W}_{zy,t}^\top = (1 - \eta)^t \left(\mathbf{W}_{yz,0} - \mathbf{W}_{zy,0}^\top\right).$$

Therefore, provided $0 < \eta < 1$, the difference decays exponentially.

### B.2 Sign-constraining the interneuron weights

In equation 16, the matrix $\mathbf{W}_{yz}$ is preceded by a positive sign and the matrix $\mathbf{W}_{zy}$ is preceded by a negative sign, which is consistent with the fact that the principal neuron-to-interneuron synapses are excitatory whereas the interneuron-to-principal neuron synapses are inhibitory. However, this interpretation is superficial because the matrices are not constrained to be non-negative, so the network can violate Dale's principle.

One approach to enforce Dale's principle is to take *projected* gradient descent steps when updating the synaptic weight matrices, which results in the offline updates

$$\mathbf{W}_{zy} \leftarrow \left[\mathbf{W}_{zy} + \eta\left(\frac{1}{T}\mathbf{Y}\mathbf{Z}^\top - \mathbf{W}_{zy}\right)\right]_+,$$
$$\mathbf{W}_{yz} \leftarrow \left[\mathbf{W}_{yz} + \eta\left(\frac{1}{T}\mathbf{Z}\mathbf{Y}^\top - \mathbf{W}_{yz}\right)\right]_+,$$

and online updates

$$\mathbf{W}_{zy} \leftarrow [\mathbf{W}_{zy} + \eta(\mathbf{y}_t\mathbf{z}_t^\top - \mathbf{W}_{zy})]_+,$$
$$\mathbf{W}_{yz} \leftarrow [\mathbf{W}_{yz} + \eta(\mathbf{z}_t\mathbf{y}_t^\top - \mathbf{W}_{yz})]_+,$$

where $[\cdot]_+$ denotes the element-wise rectification operation. This results in Algorithm 3.

---

**Algorithm 3:** A whitening network with interneurons and sign-constrained weights

> **input** centered inputs $\{\mathbf{x}_t\}$; parameters $\gamma, \eta$
> **initialize** non-negative matrices $\mathbf{W}_{yz}, \mathbf{W}_{zy}$
> **for** $t = 1, \ldots, T$ **do**
>     **repeat**
>         $\mathbf{y}_t \leftarrow \mathbf{y}_t + \gamma(\mathbf{x}_t - \mathbf{W}_{zy}\mathbf{z}_t)$
>         $\mathbf{z}_t \leftarrow \mathbf{z}_t + \gamma(\mathbf{W}_{yz}\mathbf{y}_t - \mathbf{z}_t)$
>     **until** convergence
>     $\mathbf{W}_{yz} \leftarrow [\mathbf{W}_{yz} + \eta(\mathbf{z}_t\mathbf{y}_t^\top - \mathbf{W}_{yz})]_+$
>     $\mathbf{W}_{zy} \leftarrow [\mathbf{W}_{zy} + \eta(\mathbf{y}_t\mathbf{z}_t^\top - \mathbf{W}_{zy})]_+$
> **end for**

---

In general, this modification results in a network output that does not correspond to ZCA whitening of the inputs. That being said, it is still worth comparing the performance of the network to the network with interneurons derived in section 5. In Figure 6, we compare the performance of the offline versions of Algorithms 2 and 3. We see that Algorithm 3 equilibrates at a rate that is comparable to Algorithm 2 (and therefore much faster than Algorithm 1 for large $\alpha$). In particular, this modified algorithm appears to also exhibit the accelerated dynamics due to the overparamterization of the objective.

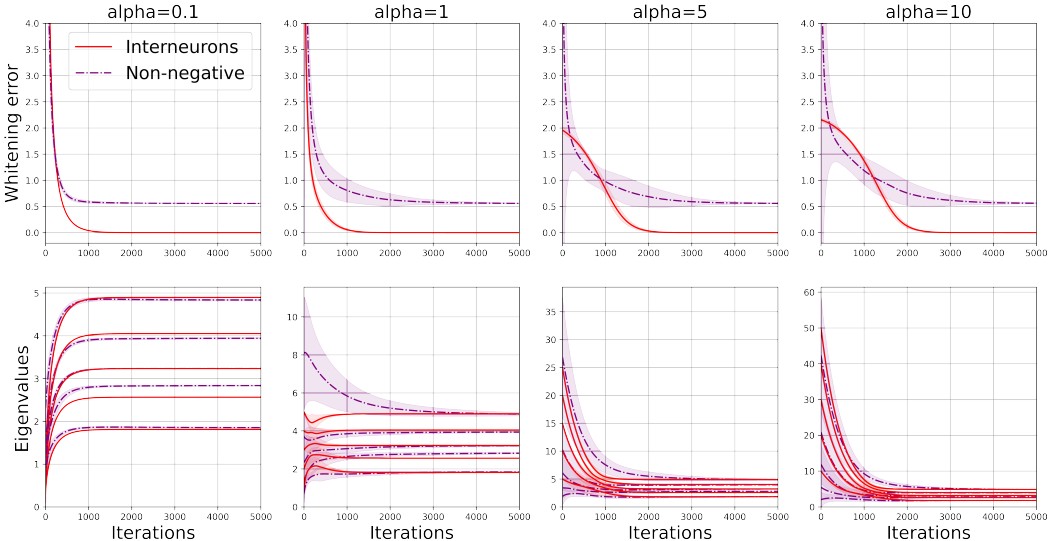

Figure 6: Comparison of whitening error and eigenvalue evolution for the offline algorithms (with non-spectral initializations) corresponding to the network with interneurons with unconstrained or non-negative weights. The lines and shaded regions respectively denote the means and 95% confidence intervals over 10 runs.

As expected, the outputs of the projected gradient descent algorithm are not fully whitened — after the synaptic dynamics of Algorithm 3 equilibrate, the eigenvalues of the output covariance $\mathbf{C}_{yy}$ are $\{1.39, 1.01, 1.00, 0.98, 0.61\}$. This can be compared with the eigenvalues of the input covariance matrix $\mathbf{C}_{xx}$: $\{24.01, 16.42, 10.45, 6.59, 3.28\}$, which suggests that the network significantly normalizes the eigenvalues of the output covariance without exactly performing ZCA whitening. In general, understanding the exact statistical transformation that results from the projected gradient descent algorithm is more mathematically challenging. For instance, in the case of Algorithm 2, the weights $\mathbf{W}$ adapt so that $\mathbf{W}\mathbf{W}^\top = \mathbf{C}_{xx}^{1/2}$; that is, the algorithm solves a symmetric matrix factorization problem whose solutions can be written in closed form in terms of the SVD of the covariance matrix $\mathbf{C}_{xx}$. In the case of Algorithm 3, the algorithm is essentially solving for the optimal (approximate) *non-negative* symmetric matrix factorization of $\mathbf{C}_{xx}^{1/2}$, for which there do not exist general closed form solutions. That being said, the empirical results suggest that the more biologically realistic network still performs rapid statistical adaptation.

### B.3 MAPPING ALGORITHM 3 ONTO THE OLFACTORY BULB

In vertebrates, an early stage of olfactory processing occurs in the olfactory bulb. The olfactory bulb receives direct olfactory inputs, which it processes and transmits to higher order brain regions (Shepherd et al., 2004), Figure 7. Olfactory receptor neurons project to the olfactory bulb, where their axon terminals cluster by receptor type into spherical structures called glomeruli. Mitral cells, which are the main projection neurons of the olfactory bulb, receive direct inputs from the olfactory receptor neuron axons and output to the rest of the brain. Each mitral cell extends its apical dendrite into a single glomerulus, where it forms synapses with the axons of the olfactory receptor neurons expressing a common receptor type. The mitral cell activities are modulated by lateral inhibition from interneurons called granule cells, which are axonless neurons that form reciprocal dendrodendritic synapses with the basal dendrites of mitral cells (Shepherd, 2009). Experimental evidence indicates that the olfactory bulb transforms its inputs so that mitral cell responses to distinct odors are approximately orthogonal (Friedrich & Laurent, 2001; 2004; Giridhar et al., 2011; Gschwend et al., 2015; Wanner & Friedrich, 2020), a transformation referred to as *pattern separation*, which is closely related to statistical whitening (Wick et al., 2010).

We explore the possibility that the mitral-granule cell microcircuit implements Algorithm 3, Figure 7. In this case, the vectors $\mathbf{x}_t$ and $\mathbf{y}_t$ represent the inputs to and outputs from the mitral cells,

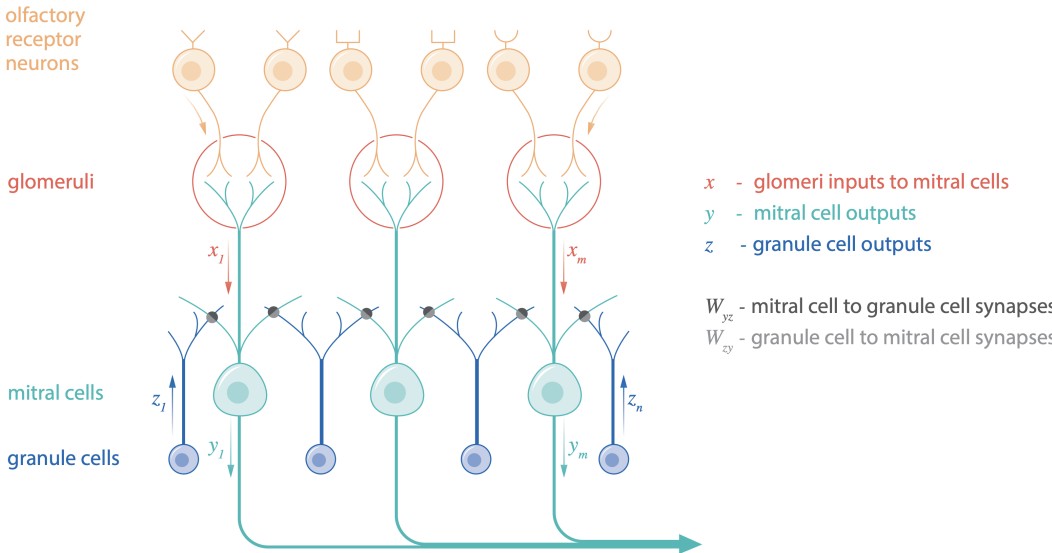

Figure 7: A simplified schematic of the olfactory bulb.

respectively. The vector $\mathbf{z}_t$ represents the granule cell outputs. We let $\mathbf{W}_{yz}$ (resp. $-\mathbf{W}_{zy}$) denote the mitral cell-to-granule cell (resp. granule cell-to-mitral cell) synaptic weights.

We compare our model with additional experimental observations. First, Algorithm 3 learns by adapting the matrices $\mathbf{W}_{yz}$ and $\mathbf{W}_{zy}$. This is in line with experimental observations that granule cells in the olfactory bulb are highly plastic suggesting that the dendrodendritic synapses are a synaptic substrate for experience dependent modulation of the olfactory bulb output (Nissant et al., 2009; Sailor et al., 2016). Second, a consequence of Algorithm 3 is that the mitral cell-to-granule cell synapses and granule cell-to-mitral cell synapses are (asymptotically) symmetric; that is, $\mathbf{W}_{zy} = \mathbf{W}_{yz}^\top$. There is experimental evidence in support of this symmetry — the dendrodendritic synapses are mainly present in reciprocal pairs (Woolf et al., 1991). In other words, the *connectivity* matrix of mitral cell-to-granule cell synapses (the matrix obtained by setting the non-zero entries in $\mathbf{W}_{yz}$ to 1) is approximately equal to the transpose of the connectivity matrix of granule cell-to-mitral cell synapses. Finally, Algorithm 3 requires that $k \geq n$; that is, there are more granule cells than mitral cells. This is consistent with measurements in the olfactory bulb indicating that there are approximately 50–100 times more granule cells than mitral cells (Shepherd et al., 2004).

## C  PROOFS OF OUR MAIN RESULTS

### C.1  PROOF OF LEMMA 1

*Proof.* Let $\sigma_1(t), \ldots, \sigma_n(t)$ be solutions of the ODE 7 and define $\mathbf{M}(t) := \mathbf{U}_x \mathbf{\Sigma}(t) \mathbf{U}_x^\top$, where $\mathbf{\Sigma}(t) := \mathrm{diag}(\sigma_1(t), \ldots, \sigma_n(t))$. Then

$$\begin{aligned}
\frac{d\mathbf{M}(t)}{dt} &= \mathbf{U}_x \mathrm{diag}\left(\frac{\lambda_1^2}{\sigma_1(t)^2}, \ldots, \frac{\lambda_n^2}{\sigma_n(t)^2}\right) \mathbf{U}_x^\top - \mathbf{I}_n \\
&= \mathbf{M}(t)^{-1} \mathbf{C}_{xx} \mathbf{M}(t)^{-1} - \mathbf{I}_n.
\end{aligned}$$

In particular, we see that $\mathbf{M}(t)$ must be the unique solution of the ODE 6, where uniqueness of solutions follows because the right-hand-side of equation 6 is locally Lipschitz continuous on its domain of definition. Equation 8 then follows from the ODE 7 and the chain rule. □

## C.2 PROOF OF PROPOSITION 1

*Proof.* Suppose $\sigma_i(0) \leq \lambda_i$ for $i = 1, \ldots, n$. By equation 8, $\sigma_i(t) \leq \lambda_i$ for all $t \geq 0$ and so

$$(\sigma_i(t)^2 - \lambda_i^2)^2 \leq (\sigma_i(0)^2 - \lambda_i^2)^2 \exp\left(-\frac{4t}{\sqrt{\lambda_i}}\right), \qquad i = 1, \ldots, n.$$

Therefore,

$$\ell(\mathbf{M}(t))^2 = \sum_{i=1}^{m} (\sigma_i(t)^2 - \lambda_i^2)^2 \leq \ell(\mathbf{M}_0)^2 \exp\left(-\frac{4t}{\sqrt{\lambda_{\max}}}\right).$$

It follows that equation 10 holds. Now suppose $\sigma_i(0) \geq \lambda_i$ for some $i = 1, \ldots, n$. By equation 7,

$$\sigma_i(t)^2 - \lambda_i^2 \geq (\sigma_i(0) - t)^2 - \lambda_i^2, \qquad t \in [0, \sigma_i(0) - \lambda_i].$$

It follows that equation 11 holds. $\qquad\square$

## C.3 PROOF OF PROPOSITION 2

*Proof.* Define $\mathbf{A}(t) := \mathbf{W}(t)\mathbf{W}(t)^\top$. By the product rule,

$$\frac{d\mathbf{A}(t)}{dt} = \mathbf{A}(t)^{-1}\mathbf{C}_{xx} + \mathbf{C}_{xx}\mathbf{A}(t)^{-1} - 2\mathbf{A}(t). \tag{17}$$

Then, by the chain rule and equation 17, for all $t \geq 0$,

$$\begin{aligned}
\frac{d\ell(\mathbf{A}(t))^2}{dt} &= 4\operatorname{Tr}\left(\left(\mathbf{A}(t)^2 - \mathbf{C}_{xx}\right)\mathbf{A}(t)\frac{d\mathbf{A}(t)}{dt}\right) \\
&= -4\operatorname{Tr}\left(\mathbf{A}(t)^2 - \mathbf{C}_{xx}\right)^2 - 4\operatorname{Tr}\left((\mathbf{A}(t)^2 - \mathbf{C}_{xx})(\mathbf{A}(t)^2 - \mathbf{A}(t)\mathbf{C}_{xx}\mathbf{A}(t)^{-1})\right) \\
&= -4\operatorname{Tr}\left(\mathbf{A}(t)^2 - \mathbf{C}_{xx}\right)^2 - 4\operatorname{Tr}\left((\mathbf{A}(t)^2 - \mathbf{C}_{xx})\mathbf{A}(t)(\mathbf{A}(t)^2 - \mathbf{C}_{xx})\mathbf{A}(t)^{-1}\right) \\
&= -4\ell(\mathbf{A}(t))^2 - 4\|\mathbf{A}(t)^{-1/2}(\mathbf{A}(t)^2 - \mathbf{C}_{xx})\mathbf{A}(t)^{1/2}\|_{\text{Frob}}^2.
\end{aligned}$$

Suppose $\mathbf{W}_0\mathbf{W}_0^\top$ is a spectral initialization. Then $\mathbf{A}(t)$ commutes with $\mathbf{C}_{xx}$ and so

$$\frac{d\ell(\mathbf{A}(t))^2}{dt} = -8\ell(\mathbf{A}(t))^2 \qquad \Rightarrow \qquad \ell(\mathbf{A}(t)) = \ell(\mathbf{A}(0))\exp(-4t).$$

Thus, equation 13 holds. For general initializations,

$$\frac{d\ell(\mathbf{A}(t))^2}{dt} \leq -4\ell(\mathbf{A}(t))^2 \qquad \Rightarrow \qquad \ell(\mathbf{A}(t)) \leq \ell(\mathbf{A}(0))\exp(-2t).$$

Therefore, inequality equation 14 holds. $\qquad\square$

