# OpenReview forum: "Interneurons accelerate learning dynamics in recurrent neural networks for statistical adaptation"
_ICLR.cc/2023/Conference — ICLR 2023 poster_

### Official Review · Reviewer_JKUS · 2022-10-22

**Confidence:** 3
**Correctness:** 3
**Technical Novelty And Significance:** 2
**Empirical Novelty And Significance:** Not applicable
**Recommendation:** 5

**Clarity, Quality, Novelty And Reproducibility:**

Writing clarity/quality:
1. Firstly, I would suggest the authors to reformulate their recurrent neural network setup to be more aligned with the convention in machine learning, given the audience of ICLR. Specifically, it is unclear how the information propagation happens in the network from the current presentation. From my understanding, the network can be thought of as receiving X as input, the input to hidden weight matrix being identity and hidden to hidden weight matrix being $-M$ (or $-WW^T$). This would simplify the neural activity dynamics to be governed by the incoming current to the neurons, $X - MY$ (or $X- WW^TY$).
2. If the above is true, can you comment on how things would change if the input to hidden weights are non-identity and/or learnable?

The work seems to target an interesting question and although there are other work that aim to understand the computational benefit of interneurons and/or inhibitory connections, I am not aware of any other work that characterized the learning dynamics in recurrent networks. So, this work presents a novel perspective.
Unfortunately, I didn't have time to try and replicate the results from the information provided by the authors. So I am unable to comment on the reproducibility.

**Strength And Weaknesses:**

Strengths:
1. The paper presents a strong theoretical analysis of synaptic change dynamics under optimization of the whitening objective. This analysis would be useful and is pertinent to current advances in self-supervised deep learning, specifically the BarlowTwins or VIC-Reg objectives (Zbontar _et al._ 2021, Bardes _et al._ 2021).
2. Through numerical simulations, the authors demonstrate that some of the assumptions in their analytical results, albeit necessary for deriving the closed form expressions are not overbearing in practice. These results suggest that the inferences hold even if commonly used initialization strategies in machine learning are adopted.

Weakness:

I have some concerns about the presentation, but I believe they can be fixed and I have described them in detail under "Clarity, Quality, Novelty and Reproducibility".

1. My main concern is from the computatinal neuroscience perspective. The authors present Dale's law as a motivating factor to understand the interneuron-mediated recurrent connections. However, it is not clear if the sign constraint is applied on the recurrent connectivity matrix or if the simulations used projected gradient descent to ensure that the weights were always all +ve. With the negative sign incorporated in the network formulation, the interneurons are supposed to be inhibitory but if the weights are allowed to be both +ve or -ve, it doesn't respect the Dale's law anymore.
2. The overparameterization narrative assumes that there are more interneurons than pyramidal neurons ($ k \geq n$), which is generally not observed in biological neural circuits. Instead it is the other way around. Could the authors comment on this and what it means for their results and inferences? Additonally, the constraint on weights being tied to and from the interneuron population, i.e. the requirement for $W$ and $W^T$, is biologically unrealistic. Could the authors comment on whether this is necessary?
3. The whitening objective provides a nice analytical setup wherein the authors converge to a closed form solution. Although the difference in dynamics for direct vs interneuron-mediated connections is presented in the paper, it is unclear how these learned representations are different, if at all. This understanding will be important for both computational neuroscience and machine learning, wherein an important question is do interneurons provide a computation benefit in realistic tasks. In other words, are interneuron-mediated recurrent networks better able to learn certain tasks because they can whiten the inputs quicker? Demonstrating this in a toy task would also provide useful evidence in favour of such an architecture. If not, it raises a question whether whitening is a reasonable objective to consider when learning representations from data.



**Summary Of The Paper:**

The authors present a theoretical analysis of the computational benefits of interneuron-mediated recurrent connections by characterizing the synaptic change dynamics while learning to whiten an input. Specifically, the authors provide analytical solutions to the dynamics of a recurrent neural network without any pointwise non-linearity when the recurrent connections are direct and when they are mediated by a layer of interneurons. They motivate the problem from a computational neuroscience and neuromorphic setting and demonstrate that interneurons enable faster synaptic plasticity dynamics, thus suggesting that it is beneficial to have an intermediate layer of interneurons in biological recurrent neural networks for faster learning. In this work, the cost function is assumed to be the whitening objective, i.e. the activity in the recurrently connected neurons are pushed towards having an Identity covariance matrix.
Although the results and insights are interesting, the work builds on certain strong assumptions that limit its utility to current studies in computational neuroscience or machine learning.

**Summary Of The Review:**

Thir work presents an interesting perspective about interneuron-mediated recurrent connections, but it uses certain biologically implausible assumptions. Moreover, the definition of recurrent network used here is slightly different from the canonical definition used by the machine learning community and given the audience of ICLR, I feel this work might not be pertinent to a broad audience. Finally, despite the interesting analytical results and corresponding empirical validation, it is unclear how the benefits in dynamics translates to computational benefits with respect to task performance or learning ability. Therefore, this work would need significant changes for it to be presentable to the ICLR community.

---

> ### Author Response · Authors · 2022-11-14
> **Response to Reviewer JKUS**
>
> Thank you for your very thoughtful and detailed review. Your comments have led us to make a number of changes which we believe substantially improve both the clarity and the impact of our submission.
>
> Thank you for pointing us to the references [Zbhontar & Jing *et al.* 2021] and [Bardes *et al.* 2021].
>
> ### Reponses to listed "Weaknesses":
>
> 1. **Sign constraint on the interneuron connectivity matrix:** We added a section (Appendix B.2) with a modified algorithm that sign-constrains the interneuron weights and still performs "rapid" statistical adaptation (but not exact ZCA whitening).
> 2. **More interneurons than principal neurons:** There are areas of the brain with more interneurons than principal cells. For example, the vertebrate olfactory bulb has approximately 50-100 times more interneurons (granule cells) than principal cells (mitral cells); see Appendix B.3 (added in the revision) and page 174 of Chapter 4 in *Synaptic Organization of the Brain* by Gordon M. Shephard (2004). Even in cases where there are fewer interneurons than principal cells, we can adapt our approach. In this case, the matrix product ${\bf WW}^\top$ is rank $k<n$ and, in general, the network will not whiten the inputs. Rather, this lower rank matrix can be used to normalize the inputs in the $k$ directions where they have highest variance (this requires modifications to the objective). Therefore, the interneurons still facilitate statistical adaptation, but not exact whitening.
>
>     **Symmetry of interneuron weights:** In Appendix B.1 (added in the revision), we decouple the interneuron weights and show that the symmetry follows asymptotically from the symmetry of the learning rules. In addition, there is biological evidence that supports the existence of symmetric weights in the vertebrate olfactory bulb (synapses occur in reciprocal pairs, see Appendix B.3).
> 3. **Are interneuron mediated recurrent networks better able to learn certain tasks because they can whiten the inputs quicker?** Yes, this is an important point. We added section 7.3 to demonstrate this on a toy example. When the inputs ${\bf x}_t$ are streamed from a fixed distribution, both the network with direct lateral connections and the network with interneurons learn to perform ZCA whitening on the inputs; that is, they learn the exact same task (albeit at different rates). However, when the inputs ${\bf x}_t$ are themselves a projection that is being learned &mdash; e.g., ${\bf x}_t={\bf V}^\top{\bf s}_t$, where ${\bf s}_t\in\mathbb{R}^d$ is activity vector of upstream inputs and ${\bf V}\in\mathbb{R}^{d\times n}$ denotes a (feedforward) synaptic weight matrix that is updated at the same time as the lateral weights &mdash; the *rate* at which the whitening transformation learns affects the *feature representation* that is learned. In particular, it is important that the whitening transform is learned sufficiently fast, so the network with interneurons learns more optimal representations. Therefore, our results are especially relevant for online learning algorithms that *simultaneously* learn the features and the whitening transformation.
>
> ### Reponses to comments on "Writing clarity/quality":
>
> 1. **Reformulation of setup to align with machine learning convention:** We have rewritten sections 3-5 to streamline the derivation of our networks and to better align with machine learning convention. In addition, immediately following each algorithm, we include a paragraph detailing its network implementation.
> 2. **How do things change if the inputs are learnable?** See our point 3 above (of our responses to "Weaknesses")
>
> [Zbhontar & Jing *et al.* 2021]: <https://proceedings.mlr.press/v139/zbontar21a.html>
> [Bardes *et al.* 2021]: <https://arxiv.org/abs/2105.04906>

---

### Official Review · Reviewer_Meay · 2022-10-25

**Confidence:** 2
**Correctness:** 4
**Technical Novelty And Significance:** 2
**Empirical Novelty And Significance:** 2
**Recommendation:** 6

**Clarity, Quality, Novelty And Reproducibility:**

Clarity
The paper is clearly written, but it would benefit from including a small section that highlights the main contributions of the work as well as the broader impact of the results.

Quality
The quality of the paper is sound. The claims in the paper are backed by solid analytical derivations and relevant numerical simulations, and I have not come across any obvious errors in them although I have not checked the math in detail.

Novelty
The finding/proposal that interneurons could make neural networks more robust against initializations is novel, but arguably, of limited relevance.

Reproducibility
The authors provide enough information to enable others to reproduce their work. I have not personally attempted to reproduce the key results of the paper.

**Strength And Weaknesses:**

Strengths
Analytical derivations and numerical simulations convincingly support the claims of the paper.

Weaknesses
Dale’s principle is a fundamental dogma in Neuroscience and thus structures that violate it cannot be deemed biologically plausible by our current standards. Comparing the advantages of networks that follow Dale’s principle against ones that don’t could be arguably deemed as little more but an interesting mathematical exercise. The main result of the paper, namely that networks obeying Dale’s principle are more robust to initializations is a bit underwhelming and likely to have very low impact in the broader field of Neuroscience and RNNs.


**Summary Of The Paper:**

The authors investigate the role of inhibitory interneurons in the statistical adaptation of the brain. More specifically the investigate their role in neural networks that perform input whitening. The authors demonstrate that networks with inhibitory interneurons are more resilient to initializations than networks with just direct recurrent connectivity that operate in violation of Dale’s principle.

**Summary Of The Review:**

A solid and interesting paper, however the impact of the findings seems a bit too limited for a broad venue such as the ICLR conference.  Maybe a more specialized venue could be a better fit for this paper. However, I am aware that I may be overlooking some aspects of the potential impact of this paper, and I will be happy to be corrected and will adjust my rating accordingly.

---

> ### Author Response · Authors · 2022-11-14
> **Response to Reviewer Meay**
>
> Thank you for your thoughtful review. We appreciate that you find our work solid and interesting and we hope that we can convince you it is of interest to the ICLR community.
>
> **Broader implications:** We recognize that the broader implications of our results may not have been clear and so, based on your suggestion, we have included a paragraph at the end of the Introduction (last paragraph of section 1) summarizing our main contributions and their potential broader implications. Here, we highlight 2 potential implications relevant to the ICLR community:
> 1. Our results suggest that adding interneurons may accelerate learning in (non-linear) RNNs trained to perform other learning tasks. In particular, we think this will be the case when adding interneurons can be viewed as overparameterizing the learning objective. We believe this result is of interest to the RNN community.
> 2. As pointed out by Reviewer JKUS, our results are relevant to recent works on self-supervised learning that use whitening/decorrelation to prevent collapse (i.e., degenerate solutions) [[1], [2], [3], [4]]. For example, our results show that overparameterized whitening objectives lead to faster learning (and more accurate learning, see section 7.3), which may be useful in the design of self-supervised learning algorithms (especially in the online setting).
>
> **Relevance to the Neuroscience community:** We also believe our analysis of the learning dynamics of recurrent networks with interneurons is relevant to the Neuroscience community (beyond an interesting mathematical exercise), even when taking Dale's Principle as fundamental dogma. By demonstrating that the network with interneurons rapidly adapts, we offer a mathematical explanation of experiments showing that neural populations in early sensory systems in the brain rapidly adapt to changing input statistics; e.g., [[5], [6], [7]].
>
> There is also recent evidence suggesting that Dale's principle does not always hold [[8]]. Granger *et al.* show that certain cortical neurons in the mouse brain release one neurotransmitter (GABA) on select targets and another neurotransmitter (acetylcholine) on a different set of targets. Therefore, biology may not be as constrained by Dale's principle as originally thought.
>
> [1]: <https://openaccess.thecvf.com/content/ICCV2021/html/Hua_On_Feature_Decorrelation_in_Self-Supervised_Learning_ICCV_2021_paper.html>
> [2]: <http://proceedings.mlr.press/v139/ermolov21a.html>
> [3]: <https://proceedings.mlr.press/v139/zbontar21a.html>
> [4]: <https://arxiv.org/abs/2105.04906>
> [5]: <https://www.nature.com/articles/nn.3382>
> [6]: <https://www.nature.com/articles/nn.4089>
> [7]: <https://www.nature.com/articles/s41593-019-0576-z>
> [8]: <https://elifesciences.org/articles/57749>

---

### Official Review · Reviewer_qLnU · 2022-10-25

**Confidence:** 4
**Correctness:** 3
**Technical Novelty And Significance:** 3
**Empirical Novelty And Significance:** Not applicable
**Recommendation:** 8

**Clarity, Quality, Novelty And Reproducibility:**

- Clarity:
The paper is well-written and the math derivations are clear.

- Novelty: the learning dynamics of a linear recurrent network should be well-established. The part of a recurrent network with interneurons seems novel.



**Strength And Weaknesses:**

The present paper is mathematically sound by theoretically analyzing the learning dynamics. I have no problem with the techniques and the results of the paper.


### Major questions
(PS: below are some of my questions regarding the results which are not a criticism of the present study.)

- The interneurons in neuroscience studies are inhibitory neurons. For the interneurons in the present model, do authors constrain the polarity of interneurons' weights? I don't find this constraint and I guess no such constraint in the study. Moreover, for the network model with interneurons, there are no direct recurrent connections between y neurons in the network (Fig. 1, right). I am curious how the result will be changed if we include direct recurrent connections.

- The recurrent connections $W$ in the network with interneurons correspond to decompose the recurrent connections $M$ in the network without interneurons, i.e., $M=WW^\top$. This implicitly assumes the connections between primary neurons y and the interneurons z are symmetric. I am happy to see some discussions in releasing this constraint in that such symmetry probably doesn't exist in the brain.

**Summary Of The Paper:**

The paper studies the learning dynamics of a linear recurrent network with and without interneurons. Theoretical analysis and numerical simulation both suggest the network with interneurons converges faster in learning.

**Summary Of The Review:**

I have gone through nearly all the math and I am certain that the math derivations are correct.

---

> ### Author Response · Authors · 2022-11-14
> **Response to Reviewer qLnU**
>
> Thank you for your careful reading of our derivations and for your positive and helpful review. We are grateful that you verified that our derivations are correct.
>
> ### Responses to "Major questions":
>
> 1. **Polarity of interneuron weights:** We added a section (Appendix B.2) with a modified algorithm that constrains the polarity of the interneuron weights. We find that the modified algorithm also exhibits accelerated convergence, suggesting that our theoretical analysis of Algorithm 2 (which is more mathematically tractable) is informative of the dynamics of the modified algorithm.
> 2. **Adding direct recurrent connections to the network with interneurons:** We could derive a network starting from the combination of the objectives in equations 2 & 3:
>     $$\min_{\bf W}\min_{\bf M}\max_{\bf Y}\left(\frac2T\text{Tr}({\bf Y}{\bf X}^\top)-\frac1T\text{Tr}[({\bf M}+{\bf WW}^\top)({\bf YY}^\top-T{\bf I}_n)]\right).$$
>     In this case, the sum ${\bf M}+{\bf WW}^\top$ acts a Lagrange multiplier that enforces the upper bound ${\bf YY}^\top\preceq T{\bf I}_n$. Following similar steps as in sections 4 and 5, we can optimize this objective by taking (stochastic) gradient descent steps would result in a network with direct lateral connections *and* interneurons. In this case, the (equilibrium) neural outputs are ${\bf y}_t=({\bf M}+{\bf WW}^\top)^{-1}{\bf x}_t$ and ${\bf z}_t={\bf W}^\top{\bf y}_t$ and the synaptic updates are the same as in Algorithms 1 and 2. However, a mathematical analysis of the gradient flow dynamics is more involved (even for spectral initializations) and it is not clear that such motifs are common in biological networks.
> 3. **Symmetry of interneuron weights:** We added a section (Appendix B.1) that decouples the interneuron weights; however, even when decoupled, the interneuron weights are *asymptotically* symmetric due to the symmetry of the learning rules. There is biological evidence that supports the existence of symmetric weights in the vertebrate olfactory bulb (i.e., synapses occur in reciprocal pairs, see Appendix B.3).

---

### Official Review · Reviewer_hapb · 2022-10-27

**Confidence:** 3
**Correctness:** 4
**Technical Novelty And Significance:** 3
**Empirical Novelty And Significance:** 3
**Recommendation:** 8

**Clarity, Quality, Novelty And Reproducibility:**

* Clarity:
The structure, wording and math is very clear.

Quality:
The theoretical derivation seems sounds, the numerical test confirm the theory.

Novelty:
The main finding is that interneurons improve the learning timescale of bioplausible implementations of Mahalanobis whitening.
The overall framework is not novel, there have been a series of papers on deep(linear network), but the implications of interneurons on learning speeds is novel to my best knowledge.

Reproducibility:
The math is fully reproducible using a pencil and paper and linear algebra. To make the numerical experiments reproducible, the authors are kindly encouraged to share their code.


**Strength And Weaknesses:**

Strength:
* Nice analytical result with biological interpretation and numerical confirmation. Very clearly written, transparent derivation.

Weaknesses:
* Not clear how that would be done by a more biologically plausible neuron using spikes.



Minor issues:
"overeparameterized" should be "overparameterized"
"Interestingly, contrast" should be "Interestingly, in contrast to"


**Summary Of The Paper:**

The paper compares two bioplausible implementations of Mahalanobis whitening based on (anti)hebbian plasticity rules, in either a network with direct connections or a network with interneurons. The core finding is that without interneurons, the convergence times scales linearly, but with interneurons it scales only logarithmic.

**Summary Of The Review:**

The submission is a nice addition to a series of paper on bioplausible implementations of basic algorithms (PCA, CCA,etc) with a plausible biological interpretation.

---

> ### Author Response · Authors · 2022-11-14
> **Response to Reviewer hapb**
>
> Thank you for your positive review! We are glad that you appeciate our analytical and numerical results and that you think that our work is clearly written with transparent derivations.
>
> Specific responses to your comments:
>
> 1. **Spiking neurons:** We agree that using spiking neurons would make our algorithm more biologically realistic. We believe this could potentially be acheived by adapting existing approaches; for example, [(Neftci *et al.* 2019)]. However, this would lead to (substantial) additional challenges in the mathematical analysis of the networks.
> 2. **Typos:** We corrected the typos you mentioned. Thank you for your careful reading of our submission.
> 3. **Code:** We have shared our code in the supplement.
>
> [(Neftci *et al.* 2019)]: <https://ieeexplore.ieee.org/abstract/document/8891809>

---

> > ### Comment · Reviewer_hapb · 2022-11-24
> > **acknowledgement of revisions and code**
> >
> > We thank for the answers and availability of the code. The updates in the manuscript and supplement further enhance the quality of the submission.

---

### Author Response · Authors · 2022-11-13
**Changes to our submission**

We thank the reviewers for their thoughtful comments and questions. We have carefully read the reviewer comments and prepared a revised version of our submission. We first describe changes to our submission and then we address general comments and concerns that were raised by the reviewers. We hope that the reviewers find these changes improve our paper and we welcome additional feedback and/or questions.

### Additions to our submission:

1. **Appendix B: Biological realism.** We modify Algorithm 2 so that it can be mapped onto a network that respects important biological constraints. In particular, we relax the weight sharing requirement (Appendix B.1) and we sign-constrain the synaptic weight matrices (Appendix B.2). In addition, we map the modified algorithm onto the vertebrate olfactory bulb, which is thought to perform a learning task closely related to statistical whitening, and show that algorithm is consistent with several experimental observations (Appendix B.3).
2. **Section 7.3: Experiment demonstrating that rapid whitening leads to more optimal representations.** Using the tractable example of Oja's PCA algorithm, we demonstrate how output whitening can prevent representation *collapse* (i.e., degenerate solutions). Further, when the whitening transform and principal projection are learned simultaneously, we show that the whitening transform must be learned (via recurrent weights) sufficiently fast relative to the principal projection (via feedforward weights) to ensure accurate/optimal representation learning. Therefore, interneurons are also useful for learning accurate/optimal representations.

### Minor changes (clarifications, organizational):

1. Based on a suggestion by Reviewer **Meay**, we added the following paragraph at the end of the Introduction (section 1) summarizing our main results and their broader implications:
> In summary, our main contribution is a theoretical and numerical analysis of the synaptic dynamics of two linear recurrent neural networks for statistical whitening &mdash; one with direct lateral connections and one with indirect lateral connections mediated by interneurons. Our analysis shows that the synaptic dynamics converge significantly faster in the network with interneurons than the network with direct lateral connections (logarithmic versus linear convergence times). Our results have potential broader implications: (i) they suggest biological interneurons may facilitate rapid statistical adaptation; (ii) including interneurons in recurrent neural networks for solving other learning tasks may also accelerate learning; (iii) overparameterized whitening objectives may be useful for developing online self-supervised learning algorithms in machine learning.
2. Based on a suggestion by Reviewer **JKUS**, we have streamlined the derivations of the networks to be more in line with machine learning convention (sections 3-5).
3. We added the following paragraph after Algorithm 1 that carefully explains how the algorithm is implemented in a recurrent neural network (we added a similar paragraph after Algorithm 2):
>Algorithm 1 can be implemented in a network with $n$ principal neurons with direct recurrent connections $-\bf M$, Figure 1 (left). At each time step $t$, the external input to the principal neurons is ${\bf x}_t$, the output of the principal neurons is ${\bf y}_t$ and the recurrent input to the principal neurons is $-{\bf My}_t$. Therefore, the total input to the principal neurons is encoded in the $n$-dimensional vector ${\bf x}_t-{\bf My}_t$. The neural outputs are updated according to the neural dynamics in Algorithm 1 until they converge at ${\bf y}_t={\bf M}^{-1}{\bf x}_t$. After the neural activities converge, the synaptic weight matrix $\bf M$ is updated according to Algorithm 1.
4. We moved the proofs of our results to Appendix C for space considerations.
5. We have included our code as supplementary material.

---

> ### Author Response · Authors · 2022-11-14
> **Responses to the general questions, comments and criticisms**
>
> ### Relevance to the ICLR audience (Reviewers **Meay** & **JKUS**)
>
> - **Implicit acceleration phenomenon in recurrent neural networks:** There are a number of highly cited works analyzing the implicit acceleration phenomon in deep linear feedforward networks trained with backpropogation; e.g., [[Saxe *et al.* 2014], [Arora *et al.* 2018], [Gidel *et al.* 2019]]. We believe that our results, which can be viewed as an analogue of the implicit acceleration phenomenon for linear recurrent neural networks trained with (anti-)Hebbian learning rules, will be of interest to an overlapping audience.
> - **Relevance to recent self-supervised learning (SSL) methods:** A number of recent SSL methods prevent collapse by whitening/decorrelating the feature representations [[Ermolov *et al.* 2021], [Zbhontar & Jing *et al.* 2021], [Bardes *et al.* 2021], [Hua & Wang *et al.* 2021]]. Therefore, as pointed out by Reviewer JKUS, Algorithm 2 is relevant and potentially useful in the design of SSL algorithms, especially online SSL algorithms in which the features and whitening transformations are learned simultaneously (see, e.g., section 7.3).
> - **Comparison to neuronal circuits:** Our work suggests a computational role for certain biological interneurons involved in statistical adaptation circuits. Further, in Appendix B.3, we show that a modification of Algorithm 2 maps onto the vertebrate olfactory bulb and is consistent with several experimental observations. Therefore, we believe our work is of interest to the computational neuroscience community, which overlaps some with the ICLR audience.
>
> ### Biological realism (Reviewers **qLnU** & **JKUS**)
>
> - **Weight sharing:** The implementation of Algorithm 2 in a network with interneurons implicity assumes the feedforward and feedback connections between the principal neurons and interneurons are symmetric. In general, enforcing such symmetry is known as the weight transport problem and is not biologically realistic. However, as we shown in Appendix B.1, we can *decouple* the feedforward and feedback weights and the symmetry (asymptotically) follows from the symmetry of the local (anti-)Hebbian learning rules. Moreover, as we discuss in Appendix B.3, there is biological evidence for this symmetry in the vertebrate olfactory bulb, a circuit that performs a transformation closely related to statistical whitening. Therefore, we do not think this is a fundamental limitation of our approach.
> - **Sign-constrained weights:** Algorithm 2 does not sign-constrain the matrix ${\bf W}$ to have non-negative entries, so the implementation of Algorithm 2 in a network with interneurons can violate Dale's principle. To address this, we consider a modification of our algorithm (Algorithm 3 in Appendix B.2) that optimizes the synaptic weight matrix by taking *projected* gradient descent steps. Algorithm 3 can be mapped onto a network with interneurons that respects Dale's principle. We test the modified algorithm on the same dataset used in section 7.1 and we find that the synaptic dynamics associated with Algorithm 3 converge quickly (i.e., on the same time scale as Algorithm 2 and therefore much faster than Algorithm 1), suggesting that the implicit acceleration phenomenon still holds in the case of the projected gradient descent algorithm. The modified algorithm normalizes the covariance of the outputs; however, it does not perform exact ZCA whitening (the exact transformation is the solution to a non-negative matrix factorization problem). In summary, our modified algorithm respects Dale's principle and "rapidly" normalizes the covariance of the outputs, which we believe supports the claim that biological interneurons facilitate rapid statistical adaptation.
>
> [Hua & Wang *et al.* 2021]: <https://openaccess.thecvf.com/content/ICCV2021/html/Hua_On_Feature_Decorrelation_in_Self-Supervised_Learning_ICCV_2021_paper.html>
> [Ermolov *et al.* 2021]: <http://proceedings.mlr.press/v139/ermolov21a.html>
> [Zbhontar & Jing *et al.* 2021]: <https://proceedings.mlr.press/v139/zbontar21a.html>
> [Bardes *et al.* 2021]: <https://arxiv.org/abs/2105.04906>
> [Saxe *et al.* 2014]: <https://arxiv.org/abs/1312.6120>
> [Arora *et al.* 2018]: <https://proceedings.mlr.press/v80/arora18a.html>
> [Gidel *et al.* 2019]: <https://papers.nips.cc/paper/2019/hash/f39ae9ff3a81f499230c4126e01f421b-Abstract.html>

---

### Decision · Program_Chairs · 2023-01-20

**Decision:**

Accept: poster

**Justification For Why Not Higher Score:**

Though this paper is technically sound, the reviewers did note that its potential impact is somewhat limited. As such, a spotlight is inappropriate.

**Justification For Why Not Lower Score:**

With a technically correct paper that has an average score of 6.75, I believe that a reject would be inappropriate.

**Metareview: Summary, Strengths And Weaknesses:**

This paper compares two different architectures of linear recurrent neural networks for data whitening, one that has direct recurrent connections, and another where recurrent interactions are mediated by interneurons. Through analysis of synaptic dynamics and simulations the authors show that the network with interneurons can converge more rapidly regardless of initialization. They provide an account of these results via the concept of over-parameterization.

The reviewers were generally positive about this paper. There was consensus that the results were technically strong, novel, and of potential interest to some readers.

There were some concerns regarding biological relevance (e.g. lack of Dale's principle being respected), how widely relevant the results are, and whether the rapid whitening leads to more optimal representations. Following rebuttals, most of the reviewers were satisfied, and the final average score was 6.75. Given this, and that there is agreement that the paper is technically sound, a decision of accept was reached.

**Note From Pc:**

if the above contains the word "oral" or "spotlight" please see: "oral" presentation means -> notable-top-5% and "spotlight" means -> notable-top-25%. As stated in our emails, we are disassociating presentation type from AC recommendations

**Summary Of Ac-Reviewer Meeting:**

N/A